# NON-ATTENTIVE TACOTRON: ROBUST AND CONTROLLABLE NEURAL TTS SYNTHESIS INCLUDING UNSUPERVISED DURATION MODELING

## ABSTRACT

This paper presents *Non-Attentive Tacotron* based on the Tacotron 2 text-to-speech model, replacing the attention mechanism with an explicit duration predictor. This improves robustness significantly as measured by unaligned duration ratio and word deletion rate, two metrics introduced in this paper for large-scale robustness evaluation using a pre-trained speech recognition model. With the use of Gaussian upsampling, Non-Attentive Tacotron achieves a 5-scale mean opinion score for naturalness of 4.41, slightly outperforming Tacotron 2. The duration predictor enables both utterance-wide and per-phoneme control of duration at inference time. When accurate target durations are scarce or unavailable in the training data, we propose a method using a fine-grained variational auto-encoder to train the duration predictor in a semi-supervised or unsupervised manner, with results almost as good as supervised training.

## 1   INTRODUCTION

Autoregressive neural text-to-speech (TTS) models using an attention mechanism are known to be able to generate speech with naturalness on par with recorded human speech. However, these types of models are known to be less robust than traditional approaches (He et al., 2019; Zheng et al., 2019; Guo et al., 2019; Battenberg et al., 2020). These autoregressive networks that predict the output one frame at a time are trained to decide whether to stop at each frame, and thus a misprediction on a single frame can result in serious failures such as early cut-off. Meanwhile, there are little to no hard constraints imposed on the attention mechanism to prevent problems such as repetition, skipping, long pause or babbling. To exacerbate the issue, these failures are rare and are thus often not represented in small test sets, such as those used in subjective listening tests. However, in customer-facing products, even a one-in-a-million chance of such problems can severely degrade the user experience.

There have been various works aimed at improving the robustness of autoregressive attention-based neural TTS models. Some of them investigated reducing the effect of the exposure bias on the autoregressive decoder, using adversarial training (Guo et al., 2019) or adding regularization to encourage the forward and backward attention to be consistent (Zheng et al., 2019). Others utilized or designed alternative attention mechanisms, such as Gaussian mixture model (GMM) attention (Graves, 2013; Skerry-Ryan et al., 2018), forward attention (Zhang et al., 2018), stepwise monotonic attention (He et al., 2019), or dynamic convolution attention (Battenberg et al., 2020). Nonetheless, these approaches do not fundamentally solve the robustness issue.

Recently, there has been a surge in the use of non-autoregressive models for TTS. Rather than predicting whether to stop on each frame, non-autoregressive models need to determine the output length ahead of time, and one way to do so is with an explicit prediction of the duration for each input token. A side benefit of such a duration predictor is that it is significantly more resilient to the failures afflicting the attention mechanism.

However, one-to-many regression problems like TTS can benefit from an autoregressive decoder as the previous mel-spectrogram frames provides context to disambiguate between multi-modal outputs.

In this paper, we propose, *Non-Attentive Tacotron*[1], a neural TTS model that combines the robust duration predictor with the autoregressive decoder of Tacotron 2 (Shen et al., 2018).

Our work is similar to DurIAN (Yu et al., 2019; Zhang et al., 2020), which incorporates the duration predictor with an autoregressive decoder. But besides the differences in architecture, we also introduce a couple of novel features in our model.

The key contributions of this paper are summarized as follows:

1. Replacing the attention mechanism in Tacotron 2 with duration prediction and upsampling modules leading to better robustness with the naturalness matching recorded natural speech;

2. Introduction of Gaussian upsampling significantly improving the naturalness compared to vanilla upsampling through repetition;

3. Global and fine-grained controlling of durations at inference time;

4. Semi-supervised and unsupervised duration modeling of Non-Attentive Tacotron, allowing the model to be trained with few to no duration annotations; and

5. More reliable evaluation metrics for measuring robustness of TTS models, as well as comparing Non-Attentive Tacotron with Tacotron 2 with respect to those metrics.

## 2 RELATED WORKS

In the past decade, model-based TTS synthesis has evolved from hidden Markov model (HMM)-based approaches (Zen et al., 2009) to using deep neural networks. Over this period, the concept of using an explicit representation of token (phoneme) durations has not been foreign. Early neural parametric synthesis models (Zen et al., 2013) require explicit alignments between input and target and include durations as part of the bag of features used to generate vocoder parameters. Explicit durations continue to be used with the advent of the end-to-end neural vocoder WaveNet (Oord et al., 2016) in works such as Deep Voice (Arik et al., 2017; Gibiansky et al., 2017) and CHiVE (Kenter et al., 2019).

As general focus turned towards end-to-end approaches, the autoregressive sequence-to-sequence model with attention mechanism used in neural machine translation (NMT) (Bahdanau et al., 2015) and automatic speech recognition (ASR) (Chan et al., 2016) became an attractive option, removing the need to represent durations explicitly. This led to works such as Char2Wav (Sotelo et al., 2017), Tacotron (Wang et al., 2017; Shen et al., 2018), Deep Voice 3 (Ping et al., 2018), and Transformer TTS (Li et al., 2019). Similar models have been used for more complicated problems, like direct speech-to-speech translation (Jia et al., 2019), speech conversion (Biadsy et al., 2019), and speech enhancement (Ding et al., 2020).

Tacotron 2, on which our work is based, is one such model. It connects a character-level encoder and an autoregressive decoder producing mel spectrogram frames with the use of a location-sensitive attention mechanism (Chorowski et al., 2015).

Recently, there has been a surge of non-autoregressive models, bringing back the use of explicit duration prediction. This approach initially surfaced in NMT (Gu et al., 2017), then made its way into TTS with models such as FastSpeech (Ren et al., 2019; 2020), AlignTTS (Zeng et al., 2020), TalkNet (Beliaev et al., 2020), and JDI-T (Lim et al., 2020). See Appendix C for a rough categorization of these models.

To train the duration predictor, FastSpeech uses target durations extracted from a pre-trained autoregressive model in teacher forcing mode, while JDI-T also extracts target durations from a separate autoregressive model but co-trains it with the feed-forward model. TalkNet uses a CTC-based ASR model to extract target durations, while CHiVE, FastSpeech 2, and DurIAN use target durations from an external aligner module utilizing forced alignment. Finally, AlignTTS forgoes target durations completely and uses a specialized alignment loss inspired by the Baum-Welch algorithm to train a mixture density network for alignment.

---

[1]Audio samples available in supplemental materials.

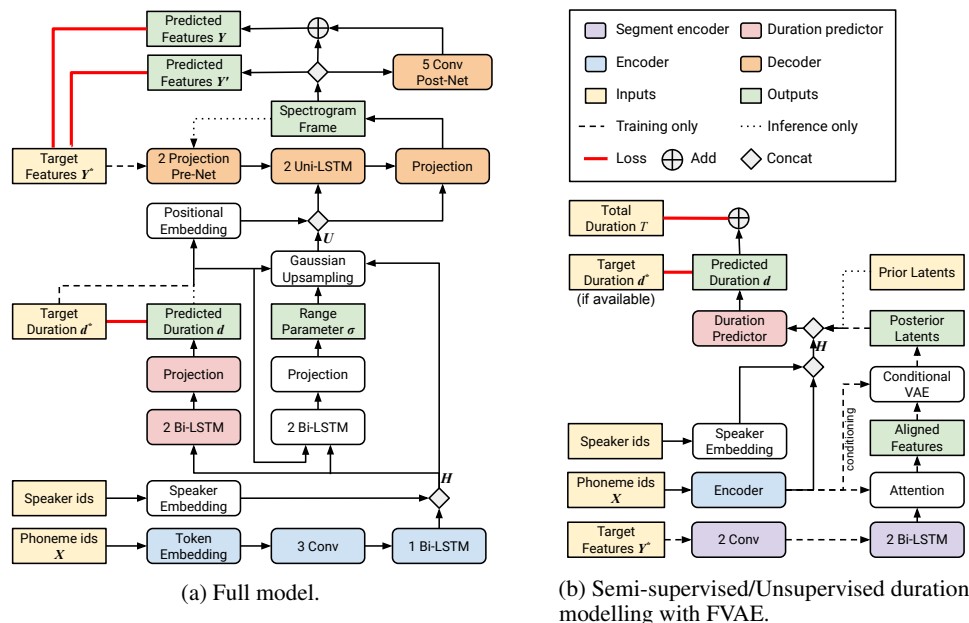

Figure 1: Architecture of Non-Attentive Tacotron.

## 3 MODEL

Modern neural TTS models typically consist of two separate networks: (1) a feature generation network that transforms input tokens (e.g., grapheme or phoneme ids) into acoustic features (e.g., mel-spectrogram), and (2) a vocoder network that transforms the acoustic features into a time-domain audio waveform. This paper focuses on the feature generation network, and can be used with any vocoder network, e.g., WaveNet (Oord et al., 2016), WaveRNN (Kalchbrenner et al., 2018), WaveGlow (Prenger et al., 2019), MelGAN (Kumar et al., 2019), or WaveGrad (Chen et al., 2020). The architecture of Non-Attentive Tacotron is illustrated in Figure 1a. See Appendix A for specific parameter value settings.

The model follows that of Tacotron 2 (Shen et al., 2018), transforming input ids $X = (x_1, \ldots, x_N)$ of length $N$ into mel-spectrogram predictions $Y = (y_1, \ldots, y_T)$ of size $T \times K$. Phonemes are used as inputs, and include a silence token at word boundaries as well as an end-of-sequence token. The ids are used to index into a learned embedding and is then passed through an encoder consisting of 3 $\times$ (dropout, batch normalization, convolution) layers followed by a single bi-directional LSTM with ZoneOut to generate a 2-dimensional output of length $N$. This output is concatenated with a speaker embedding vector to produce the final encoder output $H = (h_1, ..., h_N)$.

The autoregressive decoder also follows Tacotron 2, and predicts mel-spectrograms one frame at a time. At training time, teacher forcing (Williams & Zipser, 1989) is employed and the previous groundtruth mel-spectrogram frame is used as input, while at inference time the previous predicted mel-spectrogram frame is used. This previous frame is passed through a pre-net containing two fully-connected layers of ReLU units with dropout, then concatenated with an upsampled (aligned) encoder output corresponding to the current frame. The upsampled encoder outputs for future frames are not visible to the decoder at the current frame. In Tacotron 2, this upsampling or alignment is achieved using a location-sensitive attention mechanism (Chorowski et al., 2015), while in this work the attention mechanism is not used and a separate upsampling mechanism described later is used in its stead. The result is then passed through two uni-directional LSTM layers with ZoneOut. The LSTM output is concatenated with the upsampled encoder output yet again then projected to the mel-spectrogram dimension as frames of a preliminary predicted spectrogram $Y'$. Once all the mel-spectrogram frames have been predicted, they are passed through a 5-layer batch normalized convolutional post-net with tanh activation on all except the last layer. The post-net predicts a residual to add to the prediction $Y'$ to obtain the final prediction $Y$.

In place of the attention mechanism used in Tacotron 2, duration-based models upsample the encoder outputs using per-token duration information. This can be done by simply repeating each encoder

output by its duration as in FastSpeech (Ren et al., 2019), but instead we adopt a different process we call Gaussian upsampling, which is described in subsection 3.1. Note that while durations in seconds are used for loss computation, they are converted to durations in integer frames for upsampling.

For Gaussian upsampling, a duration and a range parameter must be predicted for each token. The range parameter is called thus because it controls the range of a token's influence. The duration predictor passes the encoder output through two bi-directional LSTM layers followed by a projection layer to predict the numeric duration $\boldsymbol{d} = (d_1, \ldots, d_N)$ for each input token. During training, these predicted durations are only used for loss computation, and the target durations are used instead in the upcoming steps[2]. The range parameter predictor passes the encoder output concatenated with durations through two bi-directional LSTM layers followed by a projection layer and a SoftPlus activation to predict a positive range parameter $\sigma$ for each input token.

After the encoder outputs are upsampled, a Transformer-style sinusoidal positional embedding (Vaswani et al., 2017) is concatenated. The positional embedding tracks the index of each upsampled frame within each token; if the duration values are $[2, 1, 3]$, the indices for the positional embedding would be $[1, 2, 1, 1, 2, 3]$.

The model is trained using a combination of duration prediction loss and mel-spectrogram reconstruction loss. The duration prediction loss is the $L^2$ loss between predicted and target durations in seconds, and the mel-spectrogram reconstruction loss is a $L^1 + L^2$ loss between the predicted and the groundtruth mel-spectrogram both before and after the post-net (following Jia et al. (2018)).

$$\mathcal{L} = \mathcal{L}_{\text{spec}} + \lambda_{\text{dur}} \mathcal{L}_{\text{dur}} \tag{1}$$

$$\mathcal{L}_{\text{dur}} = \frac{1}{N} \| \boldsymbol{d} - \boldsymbol{d}^* \|_2^2 \tag{2}$$

$$\mathcal{L}_{\text{spec}} = \frac{1}{TK} \sum_{t=1}^{T} \left( \| \boldsymbol{y}_t' - \boldsymbol{y}_t^* \|_1 + \| \boldsymbol{y}_t' - \boldsymbol{y}_t^* \|_2^2 + \| \boldsymbol{y}_t - \boldsymbol{y}_t^* \|_1 + \| \boldsymbol{y}_t - \boldsymbol{y}_t^* \|_2^2 \right) \tag{3}$$

### 3.1 GAUSSIAN UPSAMPLING

Given a sequence of vectors to be upsampled $\boldsymbol{H} = (\boldsymbol{h}_1, \ldots, \boldsymbol{h}_N)$, integer duration values $\boldsymbol{d} = (d_1, \ldots, d_N)$, and range parameter values $\boldsymbol{\sigma} = (\sigma_1, \ldots, \sigma_N)$, we compute the upsampled vector sequence $\boldsymbol{U} = (\boldsymbol{u}_1, \ldots, \boldsymbol{u}_T)$ as:

$$c_i = \frac{d_i}{2} + \sum_{j=1}^{i-1} d_j, \qquad w_{ti} = \frac{\mathcal{N}\left(t; c_i, \sigma_i^2\right)}{\sum_{j=1}^{N} \mathcal{N}\left(t; c_j, \sigma_j^2\right)}, \qquad \boldsymbol{u}_t = \sum_{i=1}^{N} w_{ti} \boldsymbol{h}_i$$

That is, we place a Gaussian distribution with standard deviation $\sigma_i$ at the center of the output segment corresponding to the $i$-th input token as determined by the duration values $\boldsymbol{d}$, and for each frame we take a weighted sum of the encoder outputs in accordance with the values of Gaussian distributions at that frame. This is similar to the softmax-based aligner in Donahue et al. (2020), except a learned $\boldsymbol{\sigma}$ rather than a fixed temperature hyperparameter is used here.

Compared with vanilla upsampling by repetition (as in Ren et al. (2019)), which can be seen as a case of learning a hard monotonic attention, Gaussian upsampling results in an alignment that is more akin to single-component GMM attention. Another benefit of Gaussian upsampling is that it is fully differentiable, which is critical to semi-supervised and unsupervised duration modeling (section 4) as it allows the gradients from the spectrogram losses to flow through to the duration predictor.

### 3.2 TARGET DURATIONS

Neural TTS models using duration need alignments between input tokens and output features. This can be accomplished by implementing an aligner module in the model or by using an external aligner.

---

[2]Note that target durations may not be required with semi-supervised and unsupervised duration modeling. See section 4.

In our work, target durations are extracted by an external, flatstart trained, speaker-dependent HMM-based aligner with a lexicon (Talkin & Wightman, 1994). However, sometimes it is difficult to train a reliable aligner model and/or extract accurate alignments due to data sparsity, poor recording conditions, or unclear pronunciations. To address this problem, we introduce semi-supervised and unsupervised duration modeling.

## 4 SEMI-SUPERVISED AND UNSUPERVISED DURATION MODELING

A naïve approach to unsupervised duration modeling would be to simply train the model using the predicted durations (instead of the target durations) for upsampling, and use only mel-spectrogram reconstruction loss for optimization. To match the length between the predicted durations and the target mel-spectrogram frames, the predicted per-token durations can be scaled by $T/\sum_i d_i$. In addition to that, an utterance-level duration loss $\mathcal{L}_{\mathrm{u}} = \frac{1}{N} \left(T - \sum_i d_i\right)^2$ could be added to the total loss. However, experiments show that such an approach does not produce satisfying naturalness in the synthesized speech (subsection 6.3).

The proposed unsupervised duration modeling is illustrated in Figure 1b. We instead utilize a fine-grained VAE (FVAE) similar to Sun et al. (2020) to model the alignment between the input tokens and the target mel-spectrogram frames, and extract per-token latent features from the target mel-spectrogram based on this alignment. The token encoder output $\boldsymbol{H}$ is aligned to the target spectrogram $\boldsymbol{Y}^*$ using an attention mechanism following Lee & Kim (2019):

$$\boldsymbol{c}_i = \mathrm{Attn}(\boldsymbol{h}_i, f_{\mathrm{spec}}(\boldsymbol{Y}^*)),$$

where $\boldsymbol{h}_i$ is used as the query in the attention, and $f_{\mathrm{spec}}$ is a spectrogram encoder whose output per frame is used as the values in the attention. A simple dot-product attention from Luong et al. (2015) was used in this work. A latent feature $\boldsymbol{z}_i$ is then computed from $\boldsymbol{c}_i$ and $\boldsymbol{h}_i$ using a variational auto-encoder (VAE) (Kingma & Welling, 2014) with a Gaussian prior $\mathcal{N}(\boldsymbol{0}, \boldsymbol{I})$, optimized through the evidence lower bound (ELBO):

$$\log p\left(\boldsymbol{Y} \mid \boldsymbol{H}\right) \geq -\sum_i D_{\mathrm{KL}}\left(q\left(\boldsymbol{z}_i \mid \boldsymbol{h}_i, \boldsymbol{c}_i\right) \| p\left(\boldsymbol{z}_i\right)\right) + \mathbb{E}_{q(\boldsymbol{z}_i|\boldsymbol{h}_i,\boldsymbol{c}_i)}\left[\log p\left(\boldsymbol{Y} \mid \boldsymbol{H}, \boldsymbol{Z}\right)\right] \quad (4)$$

where the first term is the KL divergence between the prior and posterior, and the second term can be approximated by drawing samples from the posterior.

Because these latent features are extracted from the target spectrogram with an alignment, they are capable of carrying duration related information. At training time, the per-token duration $\boldsymbol{d}$ is predicted from the concatenation of the token encoder output $\boldsymbol{H}$ and the posterior latent $\boldsymbol{Z}$; while at inference time, the prior latent is used for $\boldsymbol{Z}$ (either sampled from the distribution or using the distribution mode), and the internal attention mechanism of the FVAE is not used.

Unlike Sun et al. (2020), scheduled sampling was not utilized for factorizing latent dimensions. These latent features are only used for duration prediction, and are not used for range parameter prediction or mel-spectrogram reconstruction. We also cap the range parameters for each token to twice its predicted duration in this setup for better training stability. As in section 3, durations predicted in seconds are used for loss computation, but are converted to durations in integer frames for upsampling.

The overall loss used for semi-supervised and unsupervised training is thus

$$\mathcal{L} = \mathcal{L}_{\mathrm{spec}} + \lambda_{\mathrm{dur}}\mathcal{L}_{\mathrm{dur}} + \lambda_{\mathrm{u}}\mathcal{L}_{\mathrm{u}} + \lambda_{\mathrm{KL}}D_{\mathrm{KL}}, \quad (5)$$

where $D_{\mathrm{KL}}$ and $\mathcal{L}_{\mathrm{spec}}$ correspond to the first and second terms in Equation 4, respectively, and $\mathcal{L}_{\mathrm{dur}}$ is only counted for examples with target duration labels (i.e. supervised examples). The last three terms are all weighted per valid token.

## 5 ROBUSTNESS EVALUATION

Previous work typically evaluated the robustness of TTS systems on a small set of handpicked "hard cases" (He et al., 2019; Zheng et al., 2019; Guo et al., 2019). Although such evaluation is helpful for guiding improvements, it is not reflective of the overall robustness of the system. The handpicked

samples may be biased to the weaknesses of a certain system, and is prone to lead further optimization to overfit to the specific evaluation set.

In this work, we evaluate the robustness of TTS systems on large evaluation sets in an automated way by leveraging existing ASR systems. We run ASR and forced alignment evaluations on the synthesized speech against the verbalized text, and report two metrics measuring over- and under-generation:

1. **Unaligned duration ratio (UDR):** The synthesized speech is forced aligned with the verbalized input text using an ASR system. Each token in the input text is aligned to a segment in the synthesized audio. Any long audio segments ($> 1$ second) not aligned to any input token are typically due to over-generation from the TTS system, such as long pauses, babbling, word repetitions, or failures to stop after finishing the utterance. The total duration of such long unaligned segments divided by the total output duration is the UDR. Note that short unaligned segments are ignored. If the synthesized speech is unable to be aligned with the input text, it is considered as having a UDR of 100%.

2. **ASR word deletion rate (WDR):** This is the deletion error portion in a standard ASR word error rate (WER) evaluation. Under-generation in the synthesized speech, such as early cutoff and word skipping, is reflected by a higher WDR.

As the ASR system will make mistakes, the metrics above are just an upper-bound on the actual failures of the TTS system.

## 6 EXPERIMENTS

All models were trained on a proprietary dataset with 66 speakers with 4 different English accents (US, British, Australian, and Nigerian). The amount of data per speaker varied from merely 5 seconds to 47 hours, totaling 354 hours.

A preliminary experiment comparing different attention mechanisms (including monotonic, stepwise monotonic, dynamic convolution and GMM attention (GMMA)) showed that GMMA performed the best. We therefore compared our non-attentive Tacotron not only with Tacotron 2 with location-sensitive attention (LSA) which was used in the original Tacotron 2 paper but also with Tacotron 2 with GMMA. The Tacotron 2 models used reduction factor 2 and $L^1 + L^2$ loss.

Following Shen et al. (2018), predicted features were obtained in teacher-forcing mode from a Tacotron 2 model and used to train a WaveRNN vocoder which was then used for all experiments.

### 6.1 NATURALNESS

The naturalness of the synthesized speech was evaluated through subjective listening tests, including 5-scale Mean Opinion Score (MOS) tests and side-by-side preference tests. The sentences were synthesized using 10 US English speakers (5 male / 5 female) in a round-robin fashion. The amount of training data for the evaluated speakers varied from 3 hours to 47 hours.

Table 1 contains MOS results. Non-Attentive Tacotron with Gaussian upsampling matched Tacotron 2 (GMMA) in naturalness, and both were close to the groundtruth audio. A preference test between

Table 1: MOS with 95% confidence intervals.

| Model | MOS |
| --- | --- |
| Tacotron 2 | |
| w/ LSA | $4.35 \pm 0.05$ |
| w/ GMMA | $4.37 \pm 0.04$ |
| Non-Attentive Tacotron | |
| w/ Gauss. upsampling | $4.41 \pm 0.04$ |
| w/ vanilla upsampling | $4.13 \pm 0.05$ |
| Ground truth | $4.42 \pm 0.04$ |

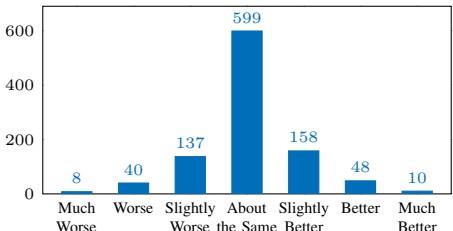

Figure 2: Preference test result with Non-Attentive Tacotron with Gaussian upsampling compared against Tacotron 2 (GMMA).

Non-Attentive Tacotron and Tacotron 2 (GMMA) further confirmed this result, as shown in Figure 2. Non-Attentive Tacotron with vanilla (repeating) upsampling was rated as significantly less natural than with Gaussian upsampling.

The effectiveness of a learned range parameter versus a fixed temperature hyperparameter set at 10.0 as per Donahue et al. (2020) is compared using a preference test in Table 2. While there is only a slight perceived benefit in using a learned range parameter, it reduces the need to tune another dataset-dependent hyperparameter. Additionally, in multi-speaker setups it is possible that the optimal $\sigma$ may be speaker-dependent.

Table 2: Preference test between a learned $\sigma$ versus a fixed $\sigma$ set at 10.0. Pace is defined as in subsection 6.2. A negative preference value means that the learned $\sigma$ is preferred over the fixed $\sigma$.

| Pace | 0.8× | 1.0× | 1.25× |
|---|---|---|---|
| Preference | $-0.017 \pm 0.057$ | $\mathbf{-0.055 \pm 0.054}$ | $-0.017 \pm 0.055$ |

## 6.2 PACE CONTROL

Table 3 shows WER and MOS results after modifying the utterance-wide pace by dividing the predicted durations by various factors. The WER is computed on speech synthesized on transcripts from the LibriTTS test-clean subset with the same 10 speakers in subsection 6.1, and then transcribed by an ASR model described in Park et al. (2020) with a WER of 2.3% on the ground truth audio.

With pace between $0.8\times - 1.25\times$, the WERs were hardly impacted. The WER was significantly worse when the pace was increased to $1.5\times$ normal, partially because the ASR model used was not optimized for speech so fast. In contrast, the subjective MOS decreased rapidly when the pace was sped up or slowed down significantly. However, most of the comments from raters were simply complaining about the pace, such as "too slow to be natural" (0.8x) or "way too fast" (1.25x).

Non-Attentive Tacotron is also able to control the pace of the synthesized speech at a finer granularity, such as per-word or per-phoneme, while still maintain the naturalness of the synthesized speech. Figure 3 shows examples of controlling the pace for specific words in a sentence.

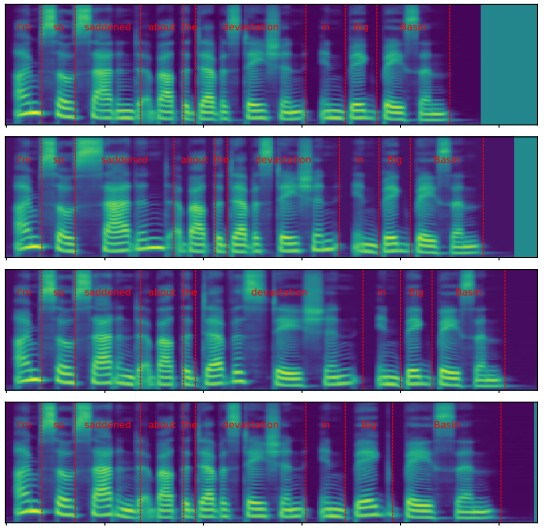

Figure 3: Single word pace control with sentence "I'm so saddened about the devastation in Big Basin." The top spectrogram is with regular pace. The rest slow down the words "saddened", "devastation", and "Big Basin" respectively to $0.67\times$ the regular pace by scaling the predicted duration by $1.5\times$.

Table 3: Performance of controlling the utterance-wide pace of the synthesized speech.

| Pace | 0.67× | 0.8× | 0.9× | 1.0× | 1.11× | 1.25× | 1.5× |
|---|---|---|---|---|---|---|---|
| WER | 3.3% | 2.8% | 2.6% | 2.6% | 2.5% | 2.7% | 6.1% |
| MOS | $3.28 \pm 0.06$ | $3.87 \pm 0.05$ | $4.24 \pm 0.04$ | $4.41 \pm 0.04$ | $4.28 \pm 0.04$ | $3.79 \pm 0.06$ | $3.18 \pm 0.06$ |

## 6.3 SEMI-SUPERVISED AND UNSUPERVISED DURATION MODELING

Ten different US English speakers (5 male / 5 female) each with about 4 hours of training data were used for evaluating the performance of the unsupervised and semi-supervised duration modeling.

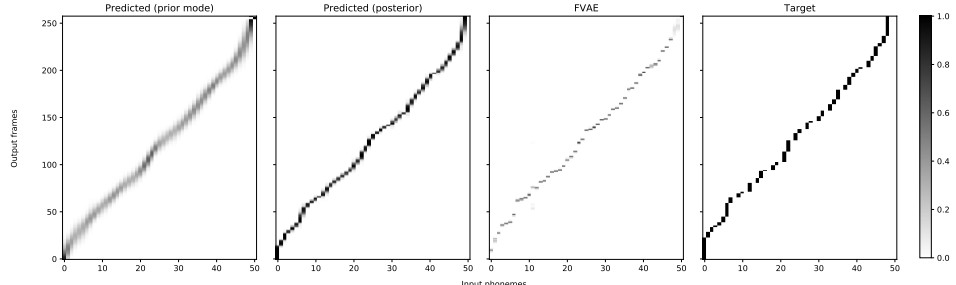

Figure 4: Alignment on text "What time do I need to show up to my sky diving lesson?" from the unsupervised model. The predicted alignments are from Gaussian upsampling.

Table 4: Performance of unsupervised and semi-supervised duration modeling. Zero vectors are used as FVAE latents for inference. MAE denotes the mean absolute error.

| Training | Model | Dur. MAE (ms) | MOS |
|---|---|---|---|
| Unsupervised | w/o FVAE | 124.4 | $2.91 \pm 0.09$ |
| | w/ FVAE | 41.3 | $4.31 \pm 0.04$ |
| Semi-supervised | w/o FVAE | 21.5 | $4.19 \pm 0.05$ |
| | w/ FVAE | 18.3 | $4.35 \pm 0.04$ |
| Supervised | Non-Attentive Tacotron | 15.4 | $4.37 \pm 0.04$ |
| | Tacotron 2 w/ GMMA | - | $4.35 \pm 0.04$ |
| Ground truth | | - | $4.52 \pm 0.03$ |

The duration labels for these 10 speakers (i.e. about 11% of the training data) were withheld for the semi-supervised models, and all duration labels were withheld for the unsupervised models.

Figure 4 shows predicted alignment after Gaussian upsampling and the internal alignment from the attention module in the FVAE compared with the alignment computed from the target durations, for the unsupervised model. Despite not having access to any target durations, both the FVAE and duration predictor were able to produce an alignment close to that computed from the target durations.

As shown in Table 4, with the use of the FVAE, the naturalness of both semi-supervised and unsupervised models were very close to that of the supervised models, even though duration prediction errors were higher. The autoregressive decoder trained with teacher forcing may have been powerful enough to correct the duration prediction errors to some degree. However, the naturalness degraded significantly without the use of the FVAE. Although the duration error from the semi-supervised model without FVAE was lower than that from the unsupervised model with FVAE, the former was significantly less natural than the latter. This may be due to a lower consistency between supervised and unsupervised speakers without FVAE.

Although these models were close to the supervised model in MOS, manual investigation found that samples from both semi-supervised models and unsupervised models had a small chance of containing slight errors that do not occur in the supervised model, such as unclear pronunciations, phoneme repetitions, or extra pauses. However, they are significantly less severe than similar errors from Tacotron 2, mostly impacting just one or a few phonemes. These errors are further confirmed in the large scale robustness evaluation (subsection 6.4).

The utterance-wide or fine-grained pace control (subsection 6.2) can be applied to the semi-supervised and unsupervised models as well. However, as the alignments are not as accurate, the synthesized speech with fine-grained pace control are not as natural as from the supervised model. The duration may be extended by simply inserting more silence, and the extended portion may include phoneme repetitions or unclear pronunciations.

Table 5: Robustness measured by UDR and WDR on two large evaluation sets. The evaluation speakers are unsupervised ones in the semi-supervised and unsupervised models.

| System | LibriTTS | | web-long | |
| --- | --- | --- | --- | --- |
| | UDR (%) | WDR (%) | UDR (%) | WDR (%) |
| Tacotron 2 | | | | |
|    w/ LSA | 16.96 | 0.4 | 46.04 | 4.4 |
|    w/ GMMA | 3.812 | 0.1 | 6.157 | 1.3 |
| Non-Attentive Tacotron | | | | |
|    Supervised | **0.005** | **0.1** | **0.011** | **1.1** |
|    Semi-supervised | 0.266 | 0.9 | 0.695 | 3.5 |
|    Unsupervised | 0.223 | 0.7 | 0.527 | 3.2 |

## 6.4 ROBUSTNESS

We evaluated the robustness of the neural TTS models by measuring UDR and WDR on two large evaluation sets: *LibriTTS*: 354K sentences from all train subsets from the LibriTTS corpus (Zen et al., 2019); and *web-long*: 100K long sentences mined from the web, which included a small amount of irregular text such as programming code. The median text lengths of the two sets were 74 and 224 characters, respectively. The input was synthesized using the same 10 speakers in subsection 6.3 in a round-robin fashion. All model outputs were capped at 120 seconds.

We used the ASR model trained on the LibriSpeech (Panayotov et al., 2015) and LibriLight (Kahn et al., 2020) corpora from Park et al. (2020) for measuring WDR, and a confidence islands-based forced alignment model (Chiu et al., 2018) for measuring UDR.

Table 5 shows the robustness metrics for Tacotron 2 and Non-Attentive Tacotron. Tacotron 2 (LSA) suffered from severe over-generation as measured by UDR, especially on long inputs. Manual investigation uncovered that they were typically long babbling or long silence, often at the end (failure to stop). It also had a high level of under-generation as measured by WDR, typically due to early cutoff. Tacotron 2 (GMMA) performed almost as well as the supervised Non-Attentive Tacotron in WDR because of its soft monotonic nature, which made end-of-sentence prediction easier. However, it still had significantly higher level of over-generation compared to Non-Attentive Tacotron, even when unsupervised or semi-supervised duration modeling is used for the latter. The robustness of semi-supervised and unsupervised Non-Attentive Tacotron is significantly worse than the supervised one. Manual investigation uncovered that the typical failure pattern is that part of the spectrogram is not correctly synthesized (often as silence, but sometimes as babbling), despite that the duration prediction seems reasonable. Such failure pattern contributes to both UDR and WDR. This indicates further improvements to be made. Even then, the semi-supervised and unsupervised Non-Attentive Tacotron still performs significantly better on over-generation compared to Tacotron 2.

In practice, we also observed that Tacotron 2 required significantly more care in data preprocessing to achieve this level of robustness, including consistent trimming of leading and trailing silences and filtering out utterances with long pauses. On the other hand, Non-Attentive Tacotron is significantly less sensitive to the data preprocessing steps.

## 7 CONCLUSIONS

This paper presented *Non-Attentive Tacotron*, showing a significant improvement in robustness compared to Tacotron 2 as measured by unaligned duration ratio and word deletion rate, while also slightly outperforming it in naturalness. This was achieved by replacing the attention mechanism in Tacotron 2 with an explicit duration predictor and Gaussian upsampling. We demonstrated the ability to control the pacing of the entire utterance as well as individual words using the duration predictor. We also described a method of modeling duration in a semi-supervised or unsupervised manner using Non-Attentive Tacotron when accurate target duration are scarce or unavailable by using a fine-grained variational auto-encoder, with results almost as good as supervised training.

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

# A  MODEL PARAMETERS

Table 6: Model parameters.

| | | |
|---|---|---|
| Common | Training mode | Synchronous |
| | Batch size (per replica) | 32 |
| | Replicas | 8 |
| | Parameter init | Xavier |
| | $L^2$ regularization | $1 \times 10^{-6}$ |
| | Learning rate | 0.001 |
| | Learning rate schedule | Linear rampup 4K steps then decay half every 50K steps. |
| | Optimizer | $\text{Adam}(0.9, 0.999, 1 \times 10^{-6})$ |
| | LSTM zone-out prob | 0.1 |
| | LSTM cell abs value cap | 10.0 |
| Inputs and Targets | Sampling rate (Hz) | 24,000 |
| | Normalize waveform | No |
| | Pre-emphasis | No |
| | Frame size (ms) | 50 |
| | Frame hop (ms) | 12.5 |
| | Windowing | Hanning |
| | FFT window size (point) | 2048 |
| | Mel channels $K$ | 128 |
| | Mel frequency lower bound (Hz) | 20 |
| | Mel frequency upper bound (Hz) | 12,000 |
| | Mel spectrogram dynamic range compression | $\log(x + 0.001)$ |
| | Token embedding dim | 512 |
| | Speaker embedding dim | 64 |
| Encoder | Conv kernel | $5 \times 1$ |
| | Conv dim | [512, 512, 512] |
| | Conv activation | [None, None, None] |
| | Conv batch norm decay | 0.999 |
| | Bi-LSTM dim | $512 \times 2$ |
| FVAE | Segment encoder conv kernel | $3 \times 1$ |
| | Segment encoder conv dim | [512, 512, 512] |
| | Segment encoder Bi-LSTM dim | $256 \times 2$ |
| | Layer norm attention inputs | Yes |
| | Latent dim | 8 projected to 16 |
| Duration Predictor | Bi-LSTM dim | $512 \times 2$ |
| | Projection activation | None |
| | $\lambda_{\text{dur}}$ supervised | 2.0 |
| | $\lambda_{\text{dur}}$ semi-supervised | 100.0 |
| | $\lambda_{\text{u}}$ semi-supervised | 100.0 |
| | $\lambda_{\text{KL}}$ semi-supervised | $1 \times 10^{-3}$ |
| | $\lambda_{\text{u}}$ unsupervised | 1.0 |
| | $\lambda_{\text{KL}}$ unsupervised | $1 \times 10^{-4}$ |
| Range Parameter Predictor | Bi-LSTM dim | $512 \times 2$ |
| | Projection activation | SoftPlus |
| Positional Embedding | Embedding dim | 32 |
| | Timestep denominator | 10,000 |
| Decoder | Pre-net dim supervised | [256, 256] |
| | Pre-net dim semi/unsupervised | [128, 128] |
| | Pre-net activation | [ReLU, ReLU] |
| | Pre-net dropout prob | [0.5, 0.5] |
| | LSTM dim | 1,024 |
| | LSTM init | $\text{uniform}(0.1)$ |
| | Projection init | $\text{uniform}(0.1)$ |
| | Post-net conv kernel | $5 \times 1$ |
| | Post-net conv dim | [512, 512, 512, 512, 128] |
| | Post-net conv activation | [tanh, tanh, tanh, tanh, None] |
| | Post-net conv init | $\text{uniform}(0.1)$ |

## B WER BREAKDOWNS IN THE ROBUSTNESS EVALUATION

Table 7: WER breakdowns in the robustness evaluation. Deletion rate (del) is the WDR in Table 5.

| System | LibriTTS | | | | web-long | | | |
| --- | --- | --- | --- | --- | --- | --- | --- | --- |
| | WER | del | ins | sub | WER | del | ins | sub |
| Tacotron 2 | | | | | | | | |
|   w/ LSA | 1.8 | 0.4 | 0.3 | 1.1 | 13.0 | 4.4 | 2.0 | 6.7 |
|   w/ GMMA | 1.7 | 0.1 | 0.1 | 1.5 | 10.1 | 1.3 | 1.3 | 7.4 |
| Non-Attentive Tacotron | | | | | | | | |
|   Supervised | 1.4 | 0.1 | 0.1 | 1.2 | 9.3 | 1.1 | 1.3 | 6.9 |
|   Semi-supervised | 3.3 | 0.9 | 0.2 | 2.2 | 14.1 | 3.5 | 1.6 | 9.0 |
|   Unsupervised | 3.5 | 0.7 | 0.3 | 2.6 | 15.3 | 3.2 | 2.0 | 10.1 |

## C CLASSIFICATION OF SOME TTS MODELS

Table 8: Classification of some TTS models into autoregressive (AR)/feed-forward (FF), RNN/Transformer/fully convolutional, and attention-based/duration-based.

| Model | Year | AR | FF | RNN | Transformer | Full Conv | Attention | Duration |
| --- | --- | --- | --- | --- | --- | --- | --- | --- |
| Deep Voice | 2017 | | ■ | ■ | | | | ■ |
| Char2Wav | 2017 | ■ | | ■ | | | ■ | |
| Tacotron | 2017 | ■ | | ■ | | | ■ | |
| Deep Voice 2 | 2017 | | ■ | ■ | | | | ■ |
| Tacotron 2 | 2018 | ■ | | ■ | | | ■ | |
| Deep Voice 3 | 2018 | ■ | | | | ■ | ■ | |
| Transformer TTS | 2019 | ■ | | | ■ | | ■ | |
| CHiVE | 2019 | | ■ | ■ | | | | ■ |
| DurIAN | 2019 | ■ | | ■ | | | | ■ |
| Fastspeech | 2019 | | ■ | | ■ | | | ■ |
| TalkNet | 2020 | | ■ | | | ■ | | ■ |
| AlignTTS | 2020 | | ■ | | ■ | | | ■ |
| JDI-T | 2020 | | ■ | | ■ | | | ■ |
| Non-Attentive Tacotron | 2020 | ■ | | ■ | | | | ■ |

