# OpenReview forum: "Non-Attentive Tacotron: Robust and controllable neural TTS synthesis including unsupervised duration modeling"
_ICLR.cc/2021/Conference — Reject_

### Official Review · AnonReviewer2 · 2020-10-23
**Nice MOS results but lacks novelty.**

**Rating:** 4
**Confidence:** 4

**Review:**

This paper proposes a new text-to-speech synthesis (TTS) framework that does not require attention mechanism at inference time. The authors suggest to use Duration Predictor to estimate the duration of each phoneme in a phoneme sequence.
The estimated duration of phonemes are upsampled by the proposed approach GaussianUpsampling, which shows significantly better listening test results than Vanilla repetition upsampling method. The authors also combine prior work, FVAE, to enable semi-supervised or unsupervised training (no duration labels). The authors show that naively predicting the duration in the unsupervised setting does not work very well, whereas combining FVAE makes it work significantly better.
Finally, the authors suggest two evaluations metrics, 1. Unaligned Duration Ratio (UDR) and 2. ASR word deletion rate (WDR), to test the robustness of TTS systems. The first metric was used to test the problem of over-generation, and the second metric was used to test a word skipping problem. The authors show that the proposed Non-attentive Tacotron model shows superior performance on both metrics compared to the attention-based Tacotron2. Lastly, the authors also show that both semi-supervised and unsupervised approach perform well as much as the supervised one in terms of MOS.

Strength:
The authors improve Tacotron2 to solve the problem of attention-based model such as over-generation and word skipping problem.  The experiments are well defined to show the robustness of Non-attentive model. It shows the robustness of Non-attentive TTS model by showing UDR and WDR. The MOS score is almost close to that of human voice, which is impressive.

Weakness:

---Lack of novelty---
While the results are impressive (e.g., MOS score) I feel like the proposed approaches lack novelty.
For example,
1)Duration prediction has been there for a while since Fastspeech1.
2)GaussianSampling is similar to EATS (https://arxiv.org/abs/2006.03575).
3)FVAE is also not proposed by the authors

---Lack of some experiments---
I assume the biggest advantage of Non-attentive Tacotron is the robustness in generation quality. If so, does the unsupervised Non-attentive Tacotron still has the advantages of less over-generation and word skipping problems? It’d have been nicer if the authors had shown UDR and WDR results in the unsupervised setting too.

---Writings---
Although using FVAE makes the unsupervised experiments successful, the insights on using FVAE to tackle this problem is not clearly stated. The authors must state why is combining FVAE approach expected to predict duration better than the naïve w/o FVAE version.  Also, I ask the authors to keep readers in mind little more when writing the paper. The authors should make the writings clearer when explaining the existing methods. For example, in Section3, the depth of explanations on FVAE is too short. Please elaborate more on how it can be formulated as conditional VAE framework.

Rating:
I consider it “not bad” paper, but I think the impact of this paper is not strong enough to pass the bar of ICLR because of the lack of novelty (as written in Weakness part). Therefore, I recommend rejection. However, I’d be happy to listen to the authors’ opinion regarding this issue.

Questions:

--- Questions on Vanilla upsampling ---
It has been shown by Fastspeech1 that the Vanilla upsampling  gives similar score to Tacotron2 model (3.84 vs 3.86). In this paper, however, it seems like the Vanilla upsampling is not working very well compare to Tacotron2 (4.13 vs 4.37). I’d like to ask the authors what the source of this difference could be.

--- Questions on training WaveRNN ---
The authors have written that WaveRNN model was trained on *predicted features*. In this case, I assume the ground truth waveform and the predicted features are *unaligned* because Tacotron2 autoregressively decodes features, which must be different to the ground truth mel-spectrogram. How could WaveRNN be trained well enough in this setting? Were the *predicted features* predicted using Teacher-forcing?

---

> ### Public Comment · ~Sharvil_Nanavati1 · 2020-11-17
> **Details lacking in paper**
>
> There's plenty of good work in this paper that other reviewers have commented on. I'll focus on areas where additional details could help readers better understand this work.
>
> - Which attention mechanism was used between the input tokens and target mel-spectrograms in your work? Sun et al. 2020 and Lee & Kim 2019 use different mechanisms.
> - Was scheduled training as described in Sun et al. 2020 used to extract energy-duration-F0 ordered latent dimensions?
>   - If so, what training schedule was used in your work? Sun et al. 2020 provide no details on this matter to serve as a baseline.
> - In the semi-supervised and unsupervised setting, are token durations represented as integers as described in sections 2 and 2.1?
> - In section 3.1, the authors state, "We also cap the range parameters to reasonable values in this setup." What reasonable values were used in your experiments?
> - In Eq. 4, should the KL divergence be added to the loss instead of subtracted? L is defined as a loss, not likelihood, in this work.

---

> > ### Author Response · Authors · 2020-11-22
> > **Reply**
> >
> > Thank you for the great feedback. Below are the itemized responses regarding each comment. We have incorporated them into the revised version.
> >
> > **Re: Which attention mechanism was used between the input tokens and target mel-spectrograms in your work? Sun et al. 2020 and Lee & Kim 2019 use different mechanisms.**
> >
> > The FVAE in this work uses a dot-product attention described in Luong et al. 2015. We have updated the text.
> >
> > **Re: Was scheduled training as described in Sun et al. 2020 used to extract energy-duration-F0 ordered latent dimensions?**
> >
> > Unlike Sun et al. 2020, scheduled sampling was not utilized for factorizing latent dimensions. These latent features are only used for duration prediction. We have clarified this in the text.
> >
> > **Re: In the semi-supervised and unsupervised setting, are token durations represented as integers as described in sections 2 and 2.1?**
> >
> > As in section 2, durations predicted in seconds are used for loss computation, but are converted to durations in integer frames for upsampling. We have clarified this in the text.
> >
> > **Re: In section 3.1, the authors state, "We also cap the range parameters to reasonable values in this setup." What reasonable values were used in your experiments?**
> >
> > We cap the range parameters for each token to twice its predicted duration for better training stability. We have clarified this in the text.
> >
> > **Re: In Eq. 4, should the KL divergence be added to the loss instead of subtracted? L is defined as a loss, not likelihood, in this work.**
> >
> > Thank you for pointing this out. We have updated the equation.

---

> ### Author Response · Authors · 2020-11-22
> **Reply**
>
> We thank the reviewer for the great feedback. Below are the itemized responses regarding each comment. We have incorporated them into the revised version.
>
> **Re: Lack of novelty--- While the results are impressive (e.g., MOS score) I feel like the proposed approaches lack novelty. For example, 1) Duration prediction has been there for a while since Fastspeech1. 2) GaussianSampling is similar to EATS (https://arxiv.org/abs/2006.03575). 3) FVAE is also not proposed by the authors**
>
> We would like to argue that:
>
> 1. As far as we are aware, our proposed model is the first duration-based model that synthesizes speech as natural as ground truth recordings from professional speakers. The performance of similar works such as FastSpeech 1 and 2 have very large gaps to ground truth (FastSpeech: 3.84 vs 4.41; FastSpeech 2: 3.83 vs 4.30; DurIAN didn’t report MOS on ground truth). It should also be noted that both FastSpeech 1 and 2 and DurIAN were evaluated on single-speaker datasets, while our proposed model was evaluated on a multi-speaker dataset.
>
> 1. As far as we are aware, our proposed unsupervised model is the first duration-based model that synthesizes high quality speech (naturalness MOS > 4.3) without using per-token duration labels. Amazingly, it achieves the same level of naturalness as the supervised model even in a multi-speaker set-up, and is very close to ground truth. As a comparison, similar works require additional labeled data and / or to be trained in a complicated manner (e.g. FastSpeech 1 and JDI-T require knowledge distillation, FastSpeech 2 requires duration, pitch and energy labels, and AlignTTS uses a complicated loss), but their performance still has a large gap to ground truth.
>
> 1. Our proposed semisupervised model is also the first proven instance of a model that can fuse high quality duration labels with examples that do not contain per-phoneme duration labels. In systems that use knowledge distillation for learning durations, as the durations extracted from the teacher model cannot be modified, it is not possible to reconcile them with accurate duration labels.
>
> 1. We would like to highlight our contribution on the robustness evaluation metrics. As acknowledged by AnonReviewer1, AnonReviewer3, and AnonReviewer4, this is a very welcoming contribution to the TTS community. While the robustness problems associated with neural TTS have been known for a while, there have been no persuasive metrics for evaluating and comparing the performance. We believe that our contribution solves this long-standing issue and would have a large impact. We plan to release the fully verbalized transcripts of the LibriTTS set to the public in order to make the evaluation easier to reproduce and the metrics more comparable with future papers. Our experiments using these metrics also revealed that duration-based models still do not guarantee perfect robustness, such as in our semi-supervised model and unsupervised models, which suggests the community directions for further works.
>
> 1. While we fully recognize the contribution of FastSpeech (2019), we would like to note that duration prediction is not new to TTS and exists long before the work of FastSpeech, such as in HMM-based parametric TTS (1999), neural parametric TTS (2013), WaveNet (2016), Deep Voice (2017), CHiVE (2019). Related discussion is included in the first paragraph of our introduction section.
>
> 1. While FVAE itself is not proposed in this work, using it for explicit duration modeling is a novel contribution from this work. We use FVAE purely for duration prediction rather than for spectrogram reconstruction as in Sun 2020, and this enables the semi-supervised and unsupervised duration modeling. Furthermore, in works such as Sun 2020, although FVAE is used for learning latents that are correlated to duration, it is unknown if those latents are able to model per-phoneme duration values or if they only capture relative speaking rate.
>
> **Re: Lack of some experiments--- I assume the biggest advantage of Non-attentive Tacotron is the robustness in generation quality. If so, does the unsupervised Non-attentive Tacotron still has the advantages of less over-generation and word skipping problems? It’d have been nicer if the authors had shown UDR and WDR results in the unsupervised setting too.**
>
> We have conducted the suggested evaluations and included them in the updated version of the paper in Table 5.
>
> A summary of the results: the semi-supervised and unsupervised models are significantly less robust than the supervised non-attentive Tacotron model, but are doing better than Tacotron 2 on over-generation issues. Manual investigation reveals that the typical error patterns are very different from Tacotron 2 (see Sec 6.4 in the updated paper), which suggests direction for future works.
>
> --- Continued in next reply due to character limit ---

---

> > ### Author Response · Authors · 2020-11-22
> > **Reply 2**
> >
> > --- Continued from previous reply ---
> >
> > **Re: Writings--- Although using FVAE makes the unsupervised experiments successful, the insights on using FVAE to tackle this problem is not clearly stated. The authors must state why does combining FVAE approach expected to predict duration better than the naïve w/o FVAE version. Also, I ask the authors to keep readers in mind little more when writing the paper. The authors should make the writings clearer when explaining the existing methods. For example, in Section3, the depth of explanations on FVAE is too short. Please elaborate more on how it can be formulated as conditional VAE framework.**
> >
> > We have revised the content accordingly, including an expansion on the unsupervised modeling and FVAE.
> >
> > **Re: Questions on Vanilla upsampling --- It has been shown by Fastspeech1 that the Vanilla upsampling gives similar score to Tacotron2 model (3.84 vs 3.86). In this paper, however, it seems like the Vanilla upsampling is not working very well compare to Tacotron2 (4.13 vs 4.37). I’d like to ask the authors what the source of this difference could be.**
> >
> > While we do not have a definitive answer to this question, there may be a few potential causes:
> >
> > 1. FastSpeech is trained using knowledge distillation, by feeding in the attention alignment and predicted spectrogram from another attention-based TTS model, which may compensate for the weaknesses in its own model components. Our model using vanilla upsampling is not trained with knowledge distillation.
> >
> > 1. The Tacotron 2 baseline in the FastSpeech paper may not be well tuned. In Table 1 in that paper, there is a large gap between Tacotron 2 and GT MOS (3.86 vs 4.41, or vs 4.00 for vocoder predictions on GT mel). However, in both the Tacotron 2 paper and our paper, the MOS from Tacotron 2 and GT are very close.
> >
> > 1. The FastSpeech paper did not use a top-notch vocoder, which may not be able to reveal the quality difference from the synthesizer models for some cases.
> >
> > We note that some other recent papers also reported that FastSpeech quality does not match Tacotron 2, such as:
> >   - _Miao, et al_, **Flow-TTS: A Non-Autoregressive Network for Text to Speech Based on Flow**, ICASSP 2020. https://ieeexplore.ieee.org/document/9054484
> >   - _Okamoto, et al_, **Transformer-Based Text-to-Speech with Weighted Forced Attention**, ICASSP 2020. https://ieeexplore.ieee.org/document/9053915
> >   - _Zeng, et al_, **AlignTTS: Efficient Feed-Forward Text-to-Speech System without Explicit Alignment**, ICASSP 2020. https://arxiv.org/pdf/2003.01950.pdf
> >   - _Guo, et al_, **DiDiSpeech: A Large Scale Mandarin Speech Corpus, 2020**, https://arxiv.org/pdf/2010.09275.pdf
> >   - _Lim, et al_, **JDI-T: Jointly trained Duration Informed Transformer for Text-To-Speech
> > without Explicit Alignment**, 2020. https://arxiv.org/pdf/2005.07799.pdf
> >
> > **Re: Questions on training WaveRNN --- The authors have written that WaveRNN model was trained on predicted features. In this case, I assume the ground truth waveform and the predicted features are unaligned because Tacotron2 autoregressively decodes features, which must be different to the ground truth mel-spectrogram. How could WaveRNN be trained well enough in this setting? Were the predicted features predicted using Teacher-forcing?**
> >
> > Indeed, features predicted using teacher-forcing were used for training the WaveRNN vocoder. We have revised the content to clarify this.

---

### Official Review · AnonReviewer4 · 2020-10-28
**Explicit duration information in TTS models leads to better robustness in terms of alignment**

**Rating:** 8
**Confidence:** 4

**Review:**

Summary:
In this paper the authors tackle the problem of alignment between input tokens and output acoustic features. The key contribution of this paper is replacing the attention mechanism of the Tacotron 2 with an explicit representation of token durations. The attention mechanism is vulnerable to issues such as pauses, repetitions, and skips, and hence using durations directly takes care of such issues. The challenge lies in obtaining the durations. The authors propose different methods toward that end.

First, they introduce a duration predictor in the Tacotron 2 model architecture which utilizes the encoder features to predict the durations. This duration predictor may be supervised if target durations are available. The authors train HMM-based aligners to obtain target durations for this method. They also introduce a fine-grained variational autoencoder (FVAE) which is a conditional VAE that model the alignment between phonemes and acoustic features to extract token-level features which are in turn used by the duration predictor to predict the durations. The FVAE may be semi-supervised or unsupervised.

The proposed non-attentive Tacotron model achieves similar naturalness scores to Tacotron 2 which is very close to ground truth naturalness. Additionally, the authors evaluate the proposed model for robustness. They propose two metrics which measure the severity of alignment issues in synthesized speech. The proposed model outperforms the attention-based Tacotron 2 for multiple datasets. The authors also demonstrate the ability to control the pace of utterances by modulating predicted durations. Finally, the authors demonstrate the ability of the semi-supervised and unsupervised models to achieve similar results to the supervised model by utilizing the FVAE duration model.

Pros:
1. The use of speech recognition-based metrics for measuring robustness is very interesting. More widespread adaptation of robustness as a criteria in addition to MOS scores is required. These objective metrics which can be computed for large datasets can be very helpful.
2. The duration modeller working in the supervised, semi-supervised, and unsupervised setting is another good feature of this model.

Cons:
1. The connection between the duration model and the full model is not very clear in the text. Why does the duration model have a section separate from the Model section, instead of a subsection? A suggestion would be to at least mention early in section 2 that ground truth target durations are not required since there is a semi-supervised duration modeller. There is a delay of 4-5 pages before it is clear that phoneme durations are not a requirement to use the non-attentive tacotron.
2. In section 5.4, the experiment setup is not very clear. Do you mean that out of the 66 speakers in the full dataset, 10 speakers' durations are removed (for the semi-supervised experiment)?

---

> ### Author Response · Authors · 2020-11-22
> **Reply**
>
> We thank the reviewer for the great feedback. Below are the itemized responses regarding each comment. We have incorporated them into the revised version.
>
> **Re: The connection between the duration model and the full model is not very clear in the text. …  A suggestion would be to at least mention early in section 2 that ground truth target durations are not required since there is a semi-supervised duration modeller.**
>
> We have modified the content accordingly. We would also like to note that not requiring duration labels is also mentioned in the last paragraph of Section 1.
>
> **Re: In section 5.4, … do you mean that out of the 66 speakers in the full dataset, 10 speakers' durations are removed (for the semi-supervised experiment)?**
>
> That is correct. We refined the text to be more clear.

---

### Official Review · AnonReviewer3 · 2020-10-29
**Similar ideas have been investigated.**

**Rating:** 5
**Confidence:** 3

**Review:**

Summary:
In this paper, the authors introduce a text-to-speech model based on Tacotron 2, called Non-Attentive Tacotron. Instead of an attention mechanism, a duration predictor is utilized to improve robustness, which is evaluated by two metrics, unaligned duration ratio (UDR) and word deletion rate(WDR). The authors propose semi-supervised and unsupervised duration modeling with a fine-grained variational auto-encoder (FVAE).

Reasons for score:

The paper is well-written. The experiment results of improvement in robustness are convincing. And the duration modeling in an unsupervised manner is appealing. However, using similar ideas to improve the robustness of end-to-end TTS has been investigated.


Pros:

-- With the use of Gaussian upsampling instead of location-sensitive attention (LSA), the results in Table 2 demonstrate the improvement of robustness compared to Tacotron 2.

-- Two metrics, UDR and WDR, focusing on TTS over-generation and under-generation, respectively, are introduced for robustness evaluation, which seems convincing and suitable for large-scale evaluation.

-- A method of unsupervised duration modeling is proposed. As shown in Table 4, with FVAE, the TTS system can achieve high MOS in an unsupervised manner when the target durations are not available.


Cons:

-- This paper mainly focuses on improving Tacotron 2 with regard to robustness. However, the similar weakness in Tacotron 2 has been investigated in DurIAN and FastSpeech, as mentioned in section 1. There should be more experiments on the comparison between Non-Attentive Tacotron and similar work, such as DurIAN, non-autoregressive FastSpeech, etc.

-- The introduced Gaussian upsampling and FVAE improve the robustness compared to Tacotron2, but similar ideas have been investigated before, as mentioned in the paper. In Donahue et al. (2020), a similar concept to Gaussian upsampling has been introduced. The authors should have the experiments verifying the benefit of a learned range parameter sigma. Besides, a similar duration modeling method that utilizes attention between target spectrogram and phone embedding also appears in Sun et al. (2020).


Questions:

-- The ratio of labeled duration data and unlabeled data in the semi-supervised experiment is unclear. The amount of training data and the ratio of the unlabeled duration should be compared in Table 4.

-- In Sun et al. (2020), besides the duration, pitch and energy are also predicted by the VAE. Can the similar method be applied to Non-Attentive Tacotron with the VAE?

---

> ### Author Response · Authors · 2020-11-22
> **Reply**
>
> We thank the reviewer for the great feedback. Below are the itemized responses regarding each comment. We have incorporated them into the revised version.
>
> **Re: Similar ideas have been investigated**
>
> We appreciate your feedback and your acknowledgement of our contributions on the unsupervised duration modeling and the proposed robustness metrics. We would additionally like to argue that:
>
> 1. As far as we are aware, our proposed model using unsupervised duration modeling is the first of its kind. Although it is composed of components from established approaches, using such a composition for solving a very challenging problem is a novel contribution. Even more, amazingly, our approach achieves the same level of naturalness as the supervised model even in a multi-speaker set-up, and is very close to ground truth. As a comparison, similar works require additional labeled data and / or to be trained in a complicated manner  (e.g. FastSpeech 1 and JDI-T require knowledge distillation, FastSpeech 2 requires duration, pitch and energy labels, and AlignTTS uses a complicated loss), but their performance still has a large gap to ground truth.
>
> 1. We would like to highlight again our contribution on the robustness evaluation metrics. We believe that our contribution solves this long-standing issue and would have a large impact. We plan to release the fully verbalized transcripts of the LibriTTS set to the public in order to make the evaluation easier to reproduce and the metrics more comparable with future papers. Our experiments using these metrics also revealed that duration-based models still do not guarantee perfect robustness, such as in our semi-supervised model and unsupervised models, which suggests the community directions for further works.
>
> **Re: There should be more experiments on the comparison between Non-Attentive Tacotron and similar work, such as DurIAN, non-autoregressive FastSpeech, etc.**
>
> While we definitely agree that such comparisons would be very helpful, it was our intention not to conduct such comparisons in our paper in order to avoid unfair comparisons. Neural TTS models are complicated models models to reproduce.  Neither FastSpeech nor DurIAN have an official release of the model code or checkpoint. Without spending as much effort on model implementation and tuning as we spent on the proposed models, such comparisons are likely to underestimate the performance of those baseline models, which we would like to avoid. Even though the FastSpeech paper graciously included a table of hyperparameters in the appendix, those experiments were conducted with a small single-speaker dataset, and there is no guarantee those hyperparameters will work as well on the large multi-speaker and multi-accent dataset we used in our experiments.
>
> We would also like to refer to the discussion with AnonReviewer2 on a related question (MOS comparison between FastSpeech and Tacotron 2) on potential confusion that may be caused by such comparisons.
>
> **Re: The authors should have the experiments verifying the benefit of a learned range parameter sigma (in Gaussian upsampling).**
>
> We have added experiments evaluating a learned range parameter sigma in Table 2. While there is only a slight perceived benefit in using a learned range parameter, it reduces the need to tune another dataset-dependent hyperparameter. Additionally, in multi-speaker setups it is quite possible that the optimal sigma will be speaker-dependent.
>
> **Re: A similar duration modeling method that utilizes attention between target spectrogram and phone embedding also appears in Sun et al. (2020).**
>
> Although FVAE is used for learning latents that are correlated to duration in Sun 2020, it is unknown if those latents can be used to model per-phoneme duration values or if they only capture relative speaking rate. In addition, the latents in Sun 2020 cannot be easily manipulated to provide precise control over individual durations.  Likewise, it is difficult to get precise timing information from the latents in Sun 2020 whereas the proposed approach with semi-supervised learning can. Such precise timing information can be important for various downstream applications such as synchronizing video and audio.
>
> **Re: The amount of training data and the ratio of the unlabeled duration should be compared in Table 4.**
>
> Thanks for your suggestion. We updated the text in the paper to include the ratio (i.e. 11% training data without duration labels for the semi-supervised model, and 100% without duration labels for the unsupervised model). All models in our experiments were trained on the same amount of recordings (i.e. the same set of waveforms and transcripts).
>
> **Re: In Sun et al. (2020), besides the duration, pitch and energy are also predicted by the VAE. Can the similar method be applied to Non-Attentive Tacotron with the VAE?**
>
> We are exploring this possibility as further work.

---

### Official Review · AnonReviewer1 · 2020-11-02

**Rating:** 6
**Confidence:** 2

**Review:**

This paper presents an approach based on the Tacotron model for speech synthesis, where the attention mechanism is replaced by a duration predictor. It also presents a short study on semi-supervised and unsupervised training. The paper also introduces two metrics to evaluate the robustness of the model. The experiments shows that the proposed model is on par with the Tacotron baselines in terms on MOS score and better in terms of the new metrics.

Pros:
- The presented approach improves upon the Tacotron model.
- The semi- and unsupervised learning capability of the system is significant.
- The new metrics are very welcome, as the evaluation tools for TTS are limited.
- The paper is clearly written.

Cons:
- The novelty is limited as it is mainly an incremental improvement to an established approach.
- The references are limited.

Detailed comments:
-  In terms of clarity, it's hard to say which part of the proposed approach is novel and which parts are from the original Tacotron. The authors should provide a brief description of the attentive Tacotron and clearly explain which parts they modified.
- A Related Work section is missing. Some related works are presented in the introduction, but I think the paper could benefit to have a separate section for that. It could also be a good spot to present the Tacotron approach.

Overall, the paper's novelty is limited, mainly due to it's incremental nature, but the unsupervised training capabilities and the new metrics are  significant, so I put it just above the acceptance threshold.

---

> ### Author Response · Authors · 2020-11-22
> **Reply**
>
> We thank the reviewer for the great feedback. Below are the itemized responses regarding each comment. We have incorporated them into the revised version.
>
> **Re: “The novelty is limited as it is mainly an incremental improvement to an established approach”, “it's hard to say which part of the proposed approach is novel and which parts are from the original Tacotron.”**
>
> We appreciate your feedback and your acknowledgement of our contributions on the unsupervised training capabilities and the newly proposed robustness metrics. We have revised our paper according to your suggestion to better describe the difference between our work and the original Tacotron 2. We would additionally like to argue that:
>
> 1. As far as we are aware, our proposed unsupervised model is the first of its kind. Although it is composed of components from established approaches, using such a composition for solving a very challenging problem is a novel contribution. Even more, amazingly, our approach achieves the same level of naturalness as the supervised model even in a multi-speaker set-up, and is very close to ground truth. As a comparison, similar works require additional labeled data and / or to be trained in a complicated manner (e.g. FastSpeech 1 and JDI-T require knowledge distillation, FastSpeech 2 requires duration, pitch and energy labels, and AlignTTS uses a complicated loss), but their performance still has a large gap to ground truth.
>
> 1. Our proposed semisupervised model is also the first proven instance of a model that can fuse high quality duration labels with examples that do not contain per-phoneme duration labels. In systems that use knowledge distillation for learning durations, as the durations extracted from the teacher model cannot be modified, it is not possible to reconcile them with accurate duration labels.
>
> 1. We would like to highlight again our contribution on the robustness evaluation metrics. While the robustness problems associated with neural TTS have been known for a while, there have been no persuasive metrics for evaluating and comparing the performance. We believe that our contribution solves this long-standing issue and would have a large impact. We plan to release the fully verbalized transcripts of the LibriTTS set to the public in order to make the evaluation easier to reproduce and the metrics more comparable with future papers. Furthermore, our experiments using these metrics also revealed that duration-based models still do not guarantee perfect robustness, such as in our semi-supervised model and unsupervised models, which suggests the community directions for further works.
>
> **Re: The references are limited.**
>
> We would appreciate pointers to references that we have missed and will definitely include them.
>
> **Re: The authors should provide a brief description of the attentive Tacotron and clearly explain which parts they modified.**
>
> We have reordered section 2 (now section 3) to first describe the parts from Tacotron 2 (encoder and decoder), and then the parts that have been modified for this work.
>
> **Re: A Related Work section is missing. Some related works are presented in the introduction, but I think the paper could benefit to have a separate section for that.**
>
> We have added a related works section as section 2.

---

### Author Response · Authors · 2020-11-22
**General update**

We have updated the paper with a new version incorporating the feedback from the reviewers. Changes include:
  - Related works has been split from the introduction into a standalone section
  - The paragraphs in the model section have been reordered to first cover the parts used in Tacotron 2, followed by modifications for this work
  - The semi-supervised and unsupervised duration modeling section has been expanded to include more details about the FVAE
  - We have added experiments evaluating a learned range parameter sigma in Table 2.
  - Robustness evaluation results for the semi-supervised and unsupervised models have been added to Table 5. In order to make the metrics comparable between supervised models and unsupervised / semi-supervised models, we also changed the speaker set used for evaluating the supervised models. We also moved the corresponding subsection to after the subsection on unsupervised / semi-supervised models for better flow.

---

### Decision · Program_Chairs · 2021-01-07
**Final Decision**

**Decision:**

Reject

**Comment:**

This paper investigates a non-attentive architecture of Tacotron 2 for TTS where the attention mechanism is replaced by a duration predictor.  The authors show that this change can significantly improve the robustness. In addition, the authors propose two evaluation metrics for TTS robustness, namely, unaligned duration ratio (UDR) and word deletion rate (WDR), which appear to be novel to the TTS community. The proposed non-attentive architecture yields good MOS scores in the experiments.

Overall, the paper is well written but the reviewers commented on the technical novelty of the work as it is essentially an improvement within the Tacotron 2 framework. There is also a lack of comparative study with other existing frameworks with similar techniques.  Although the authors put together a detailed rebuttal to address the comments, in the end the above two major concerns remain.